# Microwave Photonic Devices Based on Liquid Crystal on Silicon Technology

**Ruiqi Zheng [1] , Erwin H. W. Chan [2] , Xudong Wang [1],*, Xinhuan Feng [1] and Bai-Ou Guan [1]**

[1]   Guangdong Provincial Key Laboratory of Optical Fiber Sensing and Communications, Institute of Photonics Technology, Jinan University, Guangzhou 510632, China; ruiqizheng@stu2017.jnu.edu.cn (R.Z.); eexhfeng@gmail.com (X.F.); tguanbo@jnu.edu.cn (B.-O.G.)

[2]   College of Engineering, IT and Environment, Charles Darwin University, Darwin, NT 0909, Australia; erwin.chan@cdu.edu.au

*   Correspondence: txudong.wang@email.jnu.edu.cn; Tel.: +86-139-2643-0990

**Abstract:** This paper reviews the recent developments in microwave photonic devices based on liquid crystal on silicon (LCOS) technology. The operation principle, functions and important specifications of an LCOS based optical processor are described. Three microwave photonic devices, which are microwave photonic notch filters, phase shifters and couplers, reported in the past five years are focused on in this paper. In addition, a new multi-function signal processing structure based on amplitude and phase control functions in conjunction with a power splitting function in a commercial LCOS based optical processor is presented. It has the ability to realize multiple time -shifting operations and multiple frequency-independent phase shifting operations at the same time and control multiple RF signal amplitudes, in a single unit. The results for the new multi-function microwave photonic signal processor demonstrate multiple tunable true time delay and phase shifting operations with less than 3 dB amplitude variation over a very wide frequency range of 10 to 40 GHz.

**Keywords:** microwave photonics; optical signal processing; filters; phase shifters; true time delays

## 1. Introduction

The low-loss and large available time-bandwidth product capability of fiber optic systems make them attractive not only for signal transmission, but also for the generating, measuring and processing of microwave signals [1–4]. Combining microwave engineering and optoelectronics provides superior features of microwave photonics including immunity to electromagnetic interference (EMI), wide bandwidth capability, and light weight. Additionally, since the fractional bandwidth of a microwave signal modulated onto an optical carrier is small, these qualities are virtually independent of the microwave frequency, making photonics suitable for broadband applications. Microwave photonics brings new opportunities to revolutionize the microwave field. Its key benefits, including inherent speed and reconfigurability, have led to diverse applications such as electronic warfare, wired and wireless communications, radio astronomy, and terahertz spectroscopy and imaging, for tackling the problems that are difficult to solve or cannot be solved by conventional electronic approaches.

From a system perspective, the advantage of microwave photonics is its wide bandwidth capability, which makes it attractive for multi-function systems. This is significant in radars, electronic warfare and communication systems. Key subsystems that are required include widely tunable filters with high-resolution, wideband phase shifters, and hybrid couplers with a tunable coupling ratio and phase difference. For example, in radar systems, clutter reduction using a microwave photonic filter allows processing high-frequency signals directly in the optical domain, which is compatible with

advanced phased array radar systems that use fiber for signal transport [5]. The filter is an important element for the RF front end. There is a great deal of interest to make the filter widely tunable so that only one filter is required to achieve multi-band operation, which is very difficult to achieve using electronic approaches. Moreover, the microwave signal phase shifting operation in optical domain is of particular interest because it enables the operation over a wide bandwidth, together with a full phase shift range of 0° to 360° while keeping the amplitude of the microwave signal constant and fine tuning resolution, which are the fundamental requirements of phased array radar systems [6]. These requirements again cannot be achieved using electronic approaches. In other applications such as next-generation wireless communication systems, microwave photonic couplers are required for combining and splitting microwave signals with a tunable coupling ratio and phase difference at extremely high frequencies [7]. Since optical fiber is used as the transmission medium, it is attractive to perform high-speed and adaptive signal processing directly in the optical domain.

A number of techniques such as delay line [8,9], Sagnac loop interferometer [10], re-modulation [11], stimulated Brillouin scattering (SBS) [12,13], dual parallel modulation [14], fiber Bragg gratings (FBGs) [15] and liquid crystal on silicon (LCOS) based optical processor [16] have been reported to implement various microwave photonic devices. Amongst them, microwave photonic devices based on the LCOS technology have a number of advantages such as simple structure, robust performance, parallel signal processing and reconfigurability, which are the focus of this paper. We present three types of microwave photonic devices for notch filtering, phase shifting and coupling operations. We also present a new multi-function signal processor, which is based on using an LCOS based optical processor to realize multiple true time delay, frequency-independent phase shift and amplitude control operations for phased array antenna systems. Experimental results are presented, which demonstrate the multi-function signal processor operating over a very wide frequency range.

## 2. LCOS Based Optical Processor Operating Principle and Functions

The schematic showing the operation principle of an LCOS based optical processor is depicted in Figure 1. The input optical signal is dispersed by a diffraction grating before its spectral components hit a 2D LCOS array [17]. The LCOS array consists of a matrix of reflective liquid crystal elements. By applying voltages to these matrix elements, they can add individual phase shifts to the reflected signals. As the wavelengths are separated on the LCOS chip, each wavelength can be controlled independently and can be switched or filtered without interfering with other wavelengths. In short, an LCOS based optical processor is capable of controlling the amplitude and phase of different frequency components. There are various versions of LCOS-based optical processors. They are referred to as opto-VLSI processor [18], WaveShaper [17] and spatial light modulator [19], and have different performance. Amongst them, the WaveShaper manufactured by Finisar is the most widely used LCOS based optical processor. It has the ability to provide accurate and independent control on the amplitude and phase of the optical signal routed out from each output port with a frequency resolution of 1 GHz across the whole C band via programming the power splitting profile (PSP) file. Although the filter bandwidth and the frequency setting resolution of current commercial LCOS based optical processors are limited to 10 GHz and 1 GHz respectively, there are strong driving forces to increase the LCOS resolution. This comes from important consumer electronics applications such as TV and projection systems, 3D imaging, and defense. WaveShaper has a settling time of less than 500 ms. An LCOS based optical processor with a fast response time is demanded in many applications such as phased array antenna systems to achieve fast beam steering. Various approaches have been proposed to improve the LCOS based optical processor response time to less than 10 ms [20]. Other than resolution and response time, crosstalk, insertion loss and polarization dependent loss are also important in an LCOS based optical processor [21]. WaveShaper has a low insertion loss of less than 5 dB and a polarization dependent loss of less than 0.4 dB. Its performance is insensitive to environmental perturbation. Note that controlling the amplitude and phase of different frequency components has recently been demonstrated in a monolithic chip fabricated using 180 nm SOI CMOS technology [22].

This indicates that microwave photonic devices based on LCOS technology can be made to have a small size. They can be fabricated with other components for future communication or military applications.

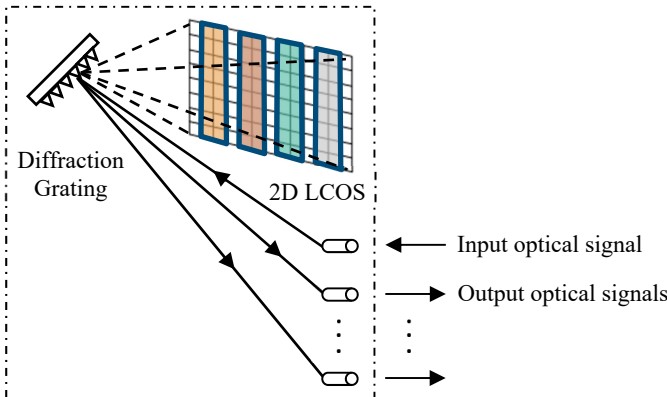

**Figure 1.** The schematic of an liquid crystal on silicon (LCOS) based optical processor.

Figure 2 shows the structure of an LCOS based microwave photonic device for realizing various signal processing operations described in the following sections. One or more optical sources, which can be telecommunication-type lasers such as DFB lasers or Fabry Perot (FP) lasers, generate continuous wave (CW) light with single or multiple wavelengths at around 1550 nm. The CW lights from the optical sources are launched into a lithium niobate optical phase modulator operated based on electro-optic effect. An RF signal is applied to the phase modulator RF input port. The output of the phase modulator consists of single- or multiple-wavelength optical carriers and pairs of anti-phase RF modulation sidebands adjacent to the carriers. An LCOS based optical processor is used to provide independent control on the amplitudes and phases of the carriers and sidebands. It can also split the processed optical signals to different output ports, which are connected to optical components, e.g., a dispersive medium to introduce different time delays for different frequency components to implement a microwave photonic filter based on a discrete time signal processing technique, before being detected by photodetectors. RF signals after signal processing are obtained from the photodetector outputs. Note that the LCOS based optical processor can be placed in any location between the phase modulator and the photodetectors to process the RF phase modulated optical signals.

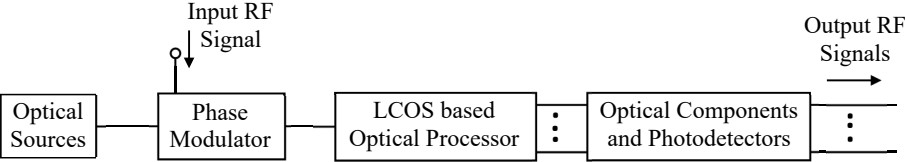

**Figure 2.** The structure of a microwave photonic device based on LCOS technology.

The performance of a microwave photonic device can be described by RF power gain, noise figure, spurious free dynamic range (SFDR) and signal-to-noise ratio (SNR) [23,24] as a fiber optic link. It is desirable for the structure shown in Figure 2 to have high power optical sources, an optical phase modulator with low insertion loss and a small switching voltage, and photodetectors with high power handling capability and high responsivity to maximize an LCOS based microwave photonic device performance. The structure shown in Figure 2 does not involve any electrical component. Hence its upper operating frequency is only limited by the bandwidth of the optical phase modulator and the photodetectors for electrical-to-optical and optical-to-electrical signal conversion process respectively. Photodetectors with a >100 GHz bandwidth are commercially available [25]. Over 100 GHz bandwidth optical phase modulators have been demonstrated [26]. However, commercial off-the-shelf optical

phase modulators have an around 65 GHz bandwidth [27], which limits the upper operating frequency of an LCOS based microwave photonic device. The lower operating frequency of an LCOS based microwave photonic device is limited by the bandwidth of the optical filter inside the LCOS based optical processor, which is 10 GHz for a commercial LCOS based optical processor (WaveShaper 4000A, Finisar, Sunnyvale, CA, USA). Techniques to extend the lower operating frequency of an LCOS based microwave photonic device down to a few gigahertz are presented in the following sections.

The use of an optical phase modulator in an LCOS based microwave photonic device has the advantage of requiring no DC bias voltage and, hence, no bias drift problem, which is present in the conventional fiber optic links that use an optical intensity modulator for RF signal modulation. An LCOS based optical processor has a very low polarization dependent loss and its performance is insensitive to changes in environmental condition. Therefore microwave photonic devices based on LCOS technology have a stable performance and hence are suitable for practical use. On the other hand, microwave photonic notch filters implemented using an FBG based delay line technique have a coherent interference problem [8] and microwave photonic phase shifters implemented using a dual-parallel Mach Zehnder modulator (DPMZM) have a bias drift problem. An LCOS based microwave photonic device does not generate additional noise components other than those in a fiber optic link, whereas using an FBG based delay line technique and SBS to implement a notch filter causes phase-induced intensity noise [28] and SBS noise [29], respectively. An LCOS based optical processor has no effect on the output RF signal nonlinearity. On the other hand, microwave photonic phase shifters implemented by controlling the bias voltages of a DPMZM or using SBS alter the output RF signal nonlinearity. This degrades the system dynamic range performance. In terms of cost, using an LCOS based optical processor to realize a single microwave signal processing operation could be expensive compared to techniques that use FBGs or a DPMZM. However, an RF modulated optical signal inside an LCOS based optical processor can be split into multiple signals with different performance characteristics, which is useful in phased array antenna systems. In this case, the overall cost of an LCOS based microwave photonic signal processor is similar or even lower than that implemented using other techniques. For example, an SBS based microwave photonic phase shifter requires a microwave signal generator to generate a pump frequency into an SBS medium to obtain an SBS effect. Four microwave signal generators are required for realizing four independent phase shifting operations, which is more expensive compared to a commercial 4-port LCOS based optical processor (WaveShaper 4000A, Finisar, Sunnyvale, CA, USA) that costs around $50K AUD. Reconfigurability is the most powerful feature of an LCOS based microwave photonic device as tuning the notch frequency of a microwave photonic filter or shifting an RF signal phase can be accomplished within a fraction of a second by uploading a definition table into an LCOS based optical processor. Furthermore, the compactness and off-the-shelf availability make an LCOS based optical processor stand out from other techniques for implementing microwave photonic devices.

## 3. Flat Passband and Narrow Notch Microwave Photonic Filters

Microwave photonic notch filters have applications, such as distributed fiber-fed antennas in defense systems [23], where specific microwave frequency components need to be eliminated. A frequency response with a flat passband and a narrow notch width is required to avoid attenuation on other frequency components around the desired null. Previous approaches to obtain a flat passband and narrow notch microwave photonic filter include techniques based on discrete time signal processing [9] and SBS [30]. The former approach can be implemented using either an infinite impulse response (IIR) or finite impulse response (FIR) delay line structure. Microwave photonic notch filter based on an IIR delay line structures have a limited free spectral range (FSR), which reduces the filter operating frequency range. Furthermore, the notch frequency cannot be tuned continuously. Unlike discrete time signal processing, the frequency response of an SBS based microwave photonic notch filter is not periodic and the notch frequency can be tuned continuously over a wide frequency range. However, the performance of an SBS based microwave photonic filter is affected by environmental perturbation

because the SBS effect is dependent on the input light polarization state. Furthermore, a recent study found that the frequency response of an SBS based microwave photonic filter is dependent on the input RF signal power [31]. In an antenna remoting application, the RF signal into a microwave photonic system is received by an antenna where the RF signal power is unknown. The RF power dependent frequency response characteristic of an SBS based microwave photonic filter prevents the filter from being used in practice. Conventional FIR delay line filters require a complex structure to realize a high-resolution frequency response. Tuning the notch frequency is difficult in a high-resolution FIR delay filter. However, the advancement of multi-wavelength optical sources and the amplitude and phase control functions in an LCOS based optical processor enable a high-resolution and continuously tunable FIR delay line notch filter to be realized using a simple structure.

The structure of a microwave photonic notch filter based on LCOS technology and the FIR delay line technique is shown in Figure 2. Here the optical source is a multi-wavelength optical source generating different-wavelength CW light into an optical phase modulator. The LCOS based optical processor connected at the output of the phase modulator has two functions in the microwave photonic notch filter. One is that it filters out one sideband of the phase modulated optical signal to obtain single sideband (SSB) modulation to avoid the dispersion-induced power fading problem [32]. Another is that it independently controls the amplitude and phase of the carriers and the remaining sidebands. The optical signals after signal processing in an LCOS based optical processor output port pass through a dispersive medium, which introduces different time delays to different-wavelength optical signals. The delayed optical signals are detected by a photodetector, which generates a notch filter shape frequency response via discrete time signal processing [8].

Each wavelength from the multi-wavelength optical source can generate one delayed optical signal or tap formed by beating between the carrier and the corresponding sideband at the photodetector. The tap amplitude is proportional to the product of the carrier and the sideband amplitude. The phase shift of the tap equals the phase difference between the carrier and the corresponding sideband. A flat passband notch filter response can be obtained by programming the LCOS based optical processor such that the taps have a sinc function distribution in the filter impulse response as illustrated in Figure 3a [33]. It can be seen from the figure that the tap distribution is symmetrical. The middle two taps have the largest and equal amplitudes, and the same phase. The next pair of taps has a 180° phase difference relative to the middle two taps. The resolution of the notch filter can be increased, which consequently reduces the notch width, by extending the number of taps in pair with this sinc function distribution. This can be obtained by increasing the number of wavelengths generated by the optical source. Techniques such as spectrum slicing a broadband optical source, multi-wavelength erbium-doped fiber laser [34], or a wideband parametric frequency comb generator can be used to generate a large number of wavelengths to realize a high-resolution notch filter response. A commercial laser array, which generates 56 different-wavelength CW light [35], can also be used as an optical source for the notch filter. Since the tap separation is proportional to the wavelength separation and the dispersion parameter of the dispersive medium, which can be made to be small, the structure can realize a large FSR frequency response. It should be noted that the ability to independently control the amplitude and the phase shift of each tap enables a highly reconfigurable notch filtering operation to be realized.

A flat passband and narrow notch microwave photonic filter can also be obtained by using a primary and secondary tap distribution impulse response as illustrated in Figure 3b [36]. In this case, the impulse response is asymmetrical and the taps have unequal separations. The amplitude of the primary tap, which is generated by a high power laser source, is equal to the sum of amplitudes of all the secondary taps. The secondary taps are generated by a multi-wavelength optical source. The secondary taps are designed to have the same time delay $\tau$ except for the first secondary tap, which needs to have $0.5\tau$ time delay to the primary tap. The time delay $\tau$ determines the filter FSR.

Thanks to the optical phase control function provided by the LCOS based optical processor, the notch frequency can be continuously tuned over the full FSR by designing the phase shift of each

tap. Note that while tuning the notch frequency, the notch width and the FSR of the notch filter remain unchanged. This shows the advantage of the optical phase control notch frequency tuning technique compared to the conventional technique that uses a wavelength tunable laser and a wavelength dependent time delay element to change the fundamental time delay of the structure, which alters not only the notch frequency but also the response shape.

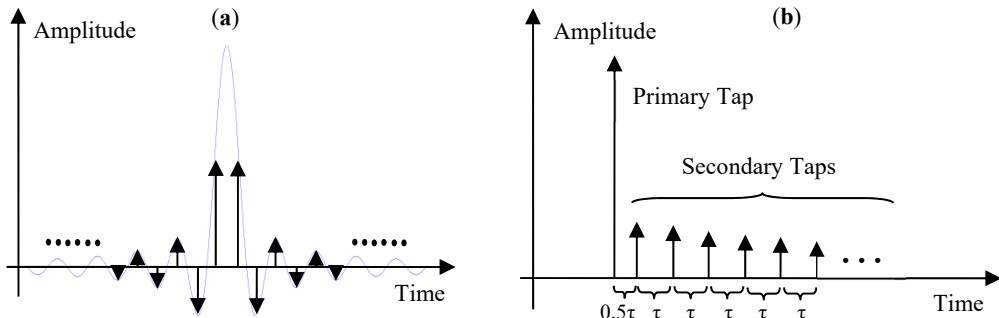

**Figure 3.** (**a**) The Sinc function distribution impulse response and (**b**) primary and secondary tap distribution impulse response microwave photonic notch filter.

An FP laser generating eight different wavelengths was used as an optical source and a 20 km standard single mode fiber was used as a dispersive medium to realize a microwave photonic notch filter with a sinc function distribution impulse response. The measured frequency response and group delay response of the microwave photonic notch filter are shown in Figure 4a. The filter −6 dB notch width was 5.1% of the −3 dB operating frequency range. The measured amplitude ripple in the notch filter passband was <1 dB. The measured notch depth was over 35 dB. The group delay response in Figure 4a shows <±25 ps ripple in the filter passband, which satisfies the requirements of radar applications [37]. Continuous notch frequency tuning was demonstrated by controlling the phases of different-wavelength optical carriers and sidebands via the LCOS based optical processor. Figure 4b shows the notch filter responses with the notch frequency tuned from 22.16 GHz to 21.84 GHz. The notch depth of >35 dB was maintained while tuning the notch frequency. The stability of the microwave photonic notch filter based on LCOS technology was investigated by measuring the eight-tap notch filter frequency response for 8 h. The eight-tap notch filter frequency response was recorded every 15 min. The results show the notch depth remains >35 dB throughout the 8-h period and the notch filter passband amplitude has a <1 dB variation. This demonstrates that the novel all-optical microwave photonic notch filter has an excellent long term stability performance, which cannot be achieved by many reported notch filtering structures. For example, the measurements in [38] show there is a more than 20 MHz change in notch frequency over 12 h in a conventional SBS based microwave photonic notch filter. The notch depth is also reduced to only around 10 dB. The change in the notch filter response shape is due to the modulator bias drift and the SBS effect is sensitive to changes in environmental conditions. An active control loop, which involves an LCOS based optical processor, is required in the SBS based microwave photonic notch filter to stabilize the notch frequency and the notch depth for long-term operation.

Continuous notch frequency tuning in the primary and secondary tap distribution impulse response microwave photonic notch filter was also demonstrated. A wavelength-tunable laser with a 16 dBm output power and an FP laser followed by an optical filter to generate six wavelengths were used as the primary and secondary optical sources respectively. The solid line in Figure 5 shows the notch filter frequency response with a notch at 23.68 GHz. The response was very stable with a notch depth of >40 dB. The amplitudes of the ripples in the notch filter passband were around 1 dB over the operating frequency range, which was 84% of the filter FSR. The figure shows that the notch frequency can be tuned to 23.29 GHz and 24.05 GHz by adjusting the phases of the secondary taps. The −6 dB

notch width, as well as the FSR, remained the same and the notch depth of >40 dB was maintained while tuning the notch frequency.

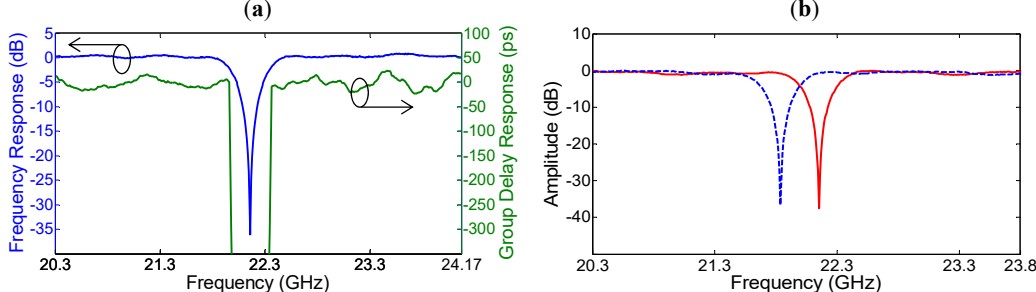

**Figure 4.** (**a**) The measured frequency response and group delay response within the notch filter −3 dB operating frequency range. (**b**) Measured tunable notch filter response with a notch frequency of 21.84 GHz (blue dashed line) and 22.16 GHz (red solid line) obtained by introducing different phase shifts to the radio frequency (RF) modulation sidebands.

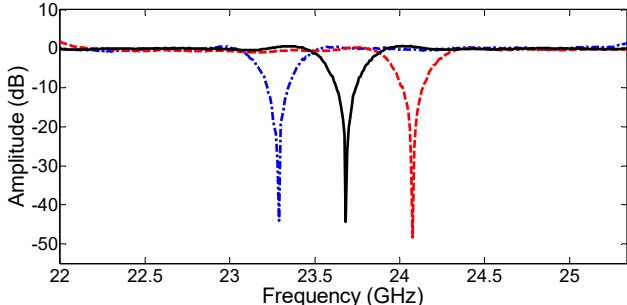

**Figure 5.** The measured tunable frequency responses of the microwave photonic notch filter with a primary and secondary tap distribution impulse response. The notch frequency is 23.29 GHz (blue dashed-dotted line), 23.68 GHz (black solid line) and 24.05 GHz (red dashed line).

## 4. Microwave Photonic Phase Shifters with Continuously Tunable 0°–360° Phase Shift

Microwave phase shifting is an important signal conditioning function that is required in signal processing for defense electronic countermeasure systems [39] and in phased-array beamforming networks for radar and satellite communication systems [40]. The phase shifter needs to provide a tunable 0°–360° phase shift and to maintain a constant amplitude in the system bandwidth. The main problem with electrical microwave phase shifters is that the phase change varies significantly depending on the frequency for a given control voltage, i.e., the phase is not flat over broad bandwidths. This introduces phase distortion to the broadband signal. Furthermore, the electrical phase shifters can only realize discrete phase shifts in a limited frequency range [41]. Moreover, it is important for the phase shifter to have a simple structure for use in multi-element phased array antenna applications [42]. Techniques based on microwave photonics can offer a solution to these limitations.

Microwave photonic phase shifters implemented using different techniques such as SBS [43], optical carrier and RF modulation sidebands amplitude and phase controls via a DPMZM [14], a polarization modulator [44], nonlinear optical loop mirrors [45] and an optical filter with a nonlinear phase response [46], have been reported. A photonic microwave phase shifter implemented using LCOS technology has the advantages of simple structure and robust performance. The power splitting function in a commercial LCOS based optical processor enables the realizing of multiple phase shifts in a single unit suitable for phased array antenna systems. An LCOS based microwave photonic phase shifter has a structure as shown in Figure 2. Phase shifting is achieved by adjusting the relative optical phase of the carrier and the sideband within the full 0° to 360° range by means of an LCOS based optical processor, and this phase shift is directly translated to the RF signal after photodetection.

The optical source generates single-wavelength CW light, which is phase modulated by an RF signal in an optical phase modulator. The RF phase modulated optical signal is processed by an LCOS based optical processor and is detected by a photodetector. The amplitude response profile of the LCOS based optical processor is designed to fix the carrier and the right sideband amplitude, as shown in Figure 6 [47], while changing the phase difference between the carrier and the right sideband to realize an RF signal phase shift. The phase difference is introduced by designing the LCOS based optical processor phase response to fix the carrier phase to 0° and to alter the right sideband phase to the desired value. Note that the left sideband at the frequency of 0 to around 20 GHz away from the optical carrier is included at the phase shifter output. The inclusion of the left sideband compensates for the unwanted effect caused by the limited LCOS resolution. This extends the LCOS based microwave photonic phase shifter operating frequency range to cover all the X, $K_u$, K, and $K_a$ bands.

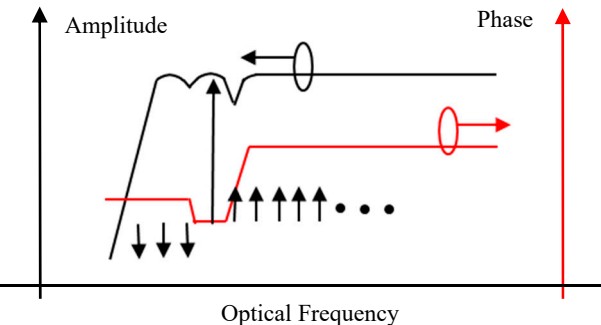

**Figure 6.** The amplitude and phase response profile of an LCOS based optical processor together with the optical carrier and the RF phase modulation sidebands showing the operation principle of the microwave photonic phase shifter based on LCOS technology.

Figure 7 shows the measured amplitude and phase response of the LCOS based microwave photonic phase shifter. The experimental results demonstrate that the structure is capable to realize a full phase shift range from −180° to 180° with <4° phase deviation over the 7.5 GHz–26.5 GHz frequency range. The amplitude response is also flat with <3 dB variation over almost 20 GHz frequency range. The maximum measurement frequency of 26.5 GHz was limited by the bandwidth of the network analyzer used in the experiment. The stability of the phase shifter was investigated by measuring the phase shifter output amplitude and phase for a 14 GHz input RF signal over a period of time. Less than 0.2 dB amplitude variation and less than 2° phase deviation were obtained over the measurement period of almost 5 h. This demonstrates that the LCOS based microwave photonic phase shifter has an excellent long term stability performance. In the case of a microwave photonic phase shifter based on controlling the bias voltages of a DPMZM, there is less than 0.5° drift in the RF signal phase in 2000 s in an open laboratory environment [48]. However, it was pointed out that performance degradation occurs in the long-term operation due to fluctuations in the environmental temperature and temperature control is required to improve long-term stability.

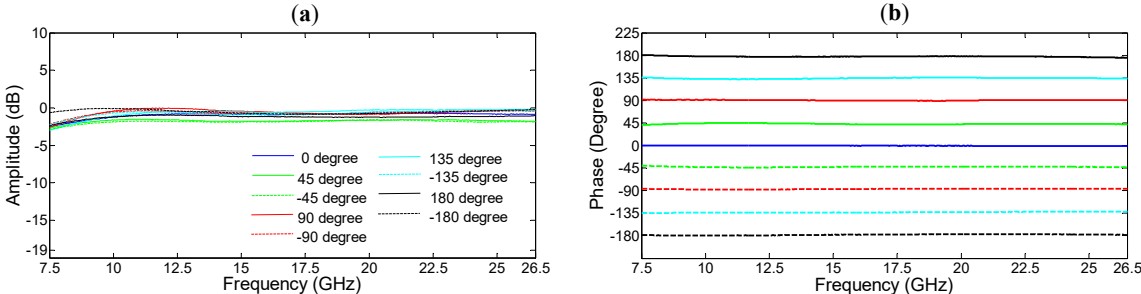

**Figure 7.** The measured (**a**) amplitude and (**b**) phase response of the microwave photonic phase shifter based on LCOS technology.

The removal of one of the RF modulation sidebands in the LCOS based microwave photonic phase shifter reduces the output RF signal amplitude. This consequently degrades the system RF power gain and SNR, which are the important performance parameters of a phase shifter [24]. Figure 8 shows another design of the LCOS amplitude and phase response profiles, which is aimed to realize a continuously tunable 0°–360° phase shift with a high SNR performance [49]. In this case, the amplitude of the optical carrier and the two RF modulation sidebands remain unchanged after the LCOS based optical processor. The optical processor phase response profile is designed so that the sum of the two beating terms, which are the left and right sidebands beat with the optical carrier at the photodetector, produces the desired RF signal phase shift. Note that the two RF modulation sidebands retain after optical signal processing. This avoids losing half of the RF information signal, which results in a higher system SNR compared to the previously reported microwave photonic phase shifter based on LCOS technology and SSB modulation. Figure 9a,b shows the LCOS based optical processor amplitude and phase response profiles for realizing a continuous −180° to 180° phase shift with a high SNR performance. The corresponding microwave photonic phase shifter output amplitude and phase responses were measured on a network analyzer and are shown in Figure 9c,d. The phase shifter output RF signal has less than 3 dB changes in amplitude for different phase shifts, and has a phase deviation of less than 5° over an 11 to 26.5 GHz frequency range. This demonstrates the frequency-independent phase shifting operation. The phase shifter output RF signal power and the noise floor were measured on an electrical spectrum analyzer connected to the photodetector output. The measurement was performed for a 20 GHz input RF signal and 0.2 modulation index. The phase shifter SNR was measured to be 139 dB in a 1 Hz bandwidth. This is around 14 dB higher than that of the previously reported phase shifter based on LCOS technology [50].

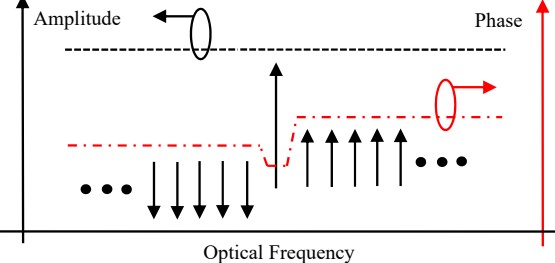

**Figure 8.** The carrier and the RF modulation sidebands at the input of the LCOS based optical processor, together with the LCOS based optical processor amplitude and phase response profile designed for improving the phase shifter signal-to-noise ratio (SNR) performance.

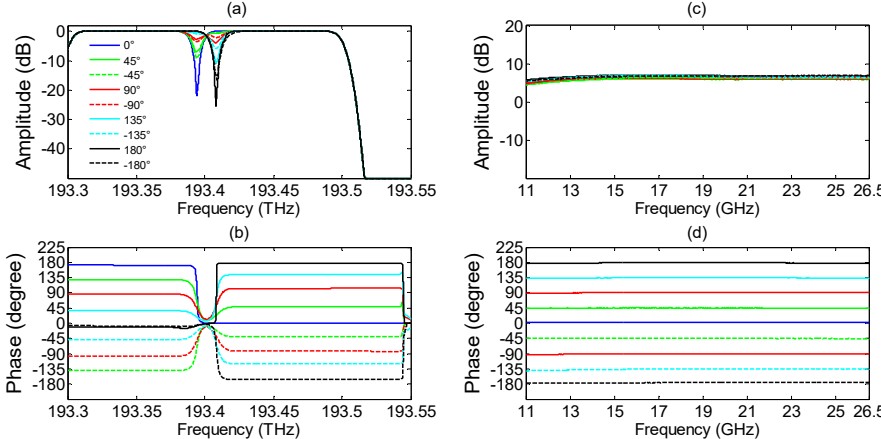

**Figure 9.** (**a**) Amplitude and (**b**) phase response profile of the LCOS based optical processor designed for improving the phase shifter SNR performance, and corresponding measured phase shifter output RF (**c**) amplitude and (**d**) phase response for different phase shifts.

## 5. Microwave Photonic Couplers with Low Amplitude and Phase Imbalance

Amongst the fundamental signal processing functions, combining two microwave signals into one and splitting a microwave signal into two or more with the desired coupling ratio and phase difference are important functions that are widely used in communication systems, radars, information processing, computing, signal analysis and measurement. Combining and splitting microwave signals are realized by electrical hybrid couplers or power dividers [51]. Broadband electrical couplers with a $0°$, $90°$ or $180°$ phase difference between the two output ports are commercially available. However, they have ripples in the amplitude and phase response. Figure 10 shows the measured frequency responses of three commercial 4 to 40 GHz bandwidth $90°$ hybrid couplers showing that the amplitude and phase imbalances are $\pm 2$ dB and $\pm 9°$ respectively. It can be seen that the ripple amplitudes increase with the frequency increases. The ripples in the phase response of a commercial 5–50 GHz bandwidth $90°$ hybrid coupler can be as high as $\pm 15°$ [52]. Ripples in the frequency response of a microwave coupler degrade the overall system performance. Photonics provides a solution to this problem.

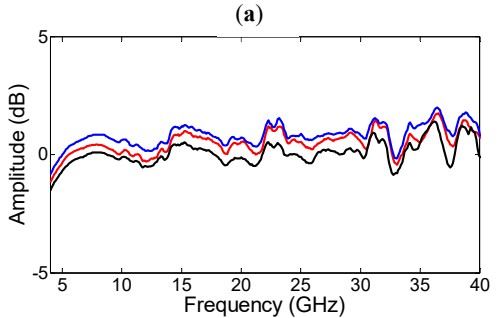 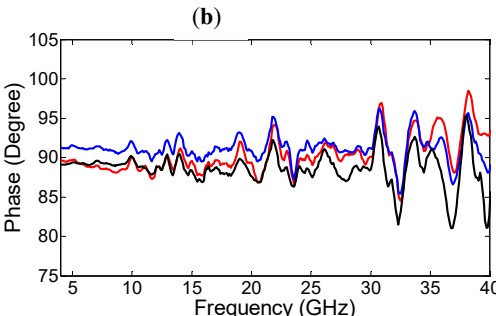

**Figure 10.** (**a**) The amplitude and (**b**) phase responses of three 4 to 40 GHz bandwidth $90°$ hybrid couplers measured at the coupler $90°$ output port with reference to the coupler $0°$ output port.

Wideband microwave photonic $90°$ hybrid couplers, which are also called Hilbert transformers, have been demonstrated using FIR microwave photonic filters [53], different types of FBGs [54], or a ring resonator-based optical all-pass filter [55]. However, they have limited bandwidth, large amplitude and phase ripples, a complex structure that requires multiple laser sources and/or modulators, or a low SNR performance. The amplitude and phase control functions together with the power splitting function in an LCOS based optical processor enables the realization of an all-optical coupler using a simple structure with very small amplitude and phase imbalances in the coupler output ports. The structure of a $90°$ hybrid coupler implemented using LCOS technology is shown in Figure 2. The principle is founded on the mapping of a photonic Hilbert transformer from the optical to the electrical domain, using spectral manipulation in the optical frequency domain by means of an LCOS based optical processor. A single-wavelength laser is used as an optical source. The LCOS based optical processor splits the RF phase modulated optical signal into two, distributes the carrier and the sidebands of the two RF phase modulated optical signals to different locations of the liquid crystal pixel and independently controls the phases of the carrier and sidebands of the two RF phase modulated optical signals. Two LCOS based optical processor output ports are connected to two photodetectors, which generate the in-phase (I-phase) and quadrature-phase (Q-phase) signals. Since the structure involves only optical components, it can operate over a very wide frequency range. The maximum operating frequency is only limited by the optical phase modulator bandwidth. Figure 11 shows the design of the LCOS based optical processor amplitude and phase response profile to minimize the loss in the system [56]. The dip in the I-phase channel of the LCOS, which is due to the abrupt transition in the optical phase, is away from the optical carrier at 193.4 THz. This avoids large attenuation in the optical signal. The small ripple in the amplitude response profile of the LCOS based optical processor Q-phase channel shown in Figure 11b is due to the change in the optical phase at around 193.4 THz. This only affects the power of the optical carrier in the Q-phase channel by a small amount. Figure 12

shows the amplitude and phase response at the I- and Q-phase output of the 90° hybrid coupler. It can be seen that the 90° hybrid coupler can operate over a wide 3 dB operating frequency range from 10.5 to 26.5 GHz. The experimental results demonstrate an amplitude imbalance of <±0.3 dB and a phase imbalance of <±0.15° over the 16 GHz bandwidth. The SNR measured at both the I and Q-phase output are above 121 dB for a 1 Hz noise bandwidth.

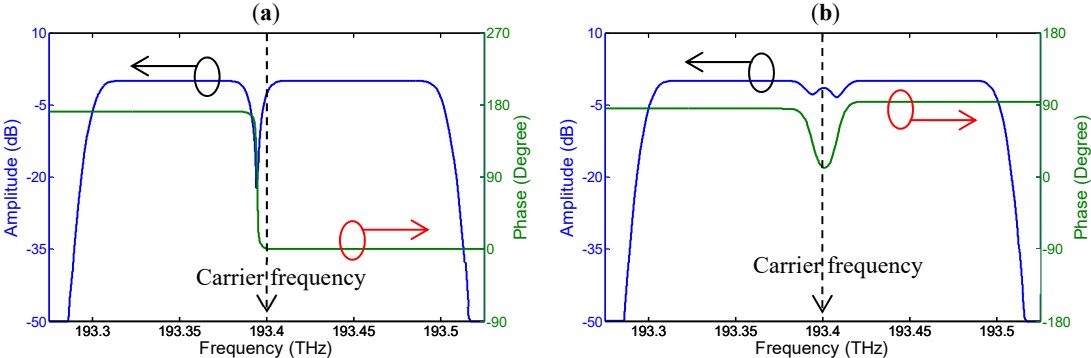

**Figure 11.** The amplitude and phase response profile of the LCOS based optical processor designed for the 90° hybrid coupler (**a**) I-phase channel and (**b**) Q-phase channel.

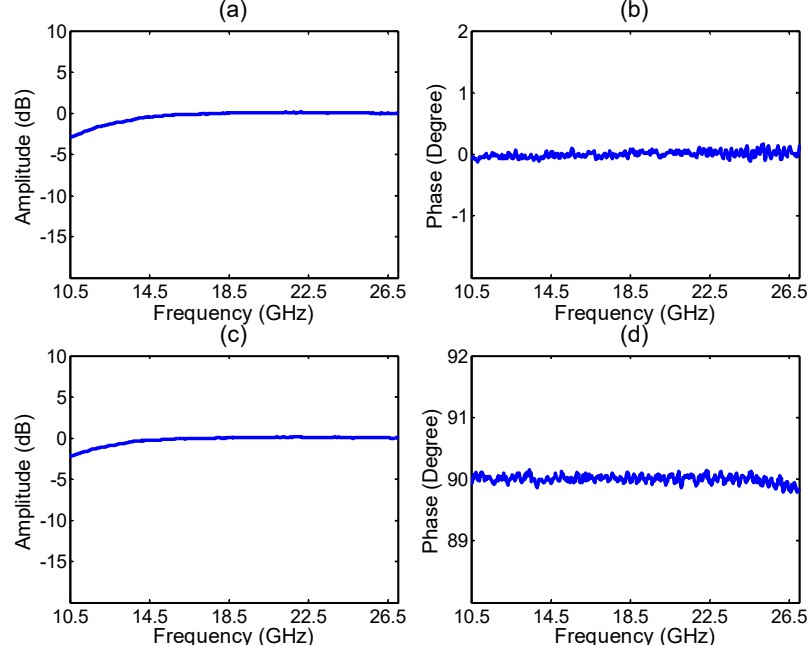

**Figure 12.** (**a**) The amplitude and (**b**) phase response of the I-phase channel, and (**c**) amplitude and (**d**) phase response of the Q-phase channel of the 90° hybrid coupler.

The idea of using an LCOS based optical processor to implement a 90° hybrid coupler can be extended to realize a 4-way RF hybrid splitter, which is also known as a quadrifilar. A 4-way hybrid splitter is used to split an RF signal into four with a phase difference of 0°, 90°, 180° and 270°. It has applications in the circularly polarized square slot antenna array [57] and the four-arm spiral antenna in a beamforming network [58]. A broadband 4-way hybrid splitter formed by connecting the two outputs of a broadband 90° hybrid coupler with two broadband 180° power splitters can have a phase error as large as ±20°. This problem can be solved by a photonics-based 4-way RF hybrid splitter implemented using LCOS technology, which has a structure as shown in Figure 2 [59]. Only a single-wavelength laser source is used and the optical phase modulator is driven by an RF signal as the 90° hybrid coupler implemented using LCOS technology. Here four LCOS based optical processor

output ports are connected to four photodetectors. The LCOS based optical processor amplitude and phase response profile for the 0° and 90° channel of the 4-way RF hybrid splitter are the same as that shown in Figure 11. The amplitude and phase response profiles for the 180° and 270° channels are designed based on the same manner as the 0° and 90° channels. Figure 13 demonstrates the function of the 4-way RF hybrid splitter using LCOS technology. It can be seen from the figure that the 4-way RF hybrid splitter can be operated over a wide 3 dB operating frequency range from 10.5 GHz to 26.5 GHz. The phase errors for all the four channels were measured and was found to be <±0.35° over the 10.5–26.5 GHz frequency range. This demonstrates that the 4-way RF hybrid splitter has an excellent phase error performance compared to its electronic counterparts. The 4-way RF hybrid splitter 0° channel was calibrated as a reference to measure the amplitude imbalance for the other three channels. It was found that there is a <1 dB amplitude imbalance in all channels over the 10.5–26.5 frequency range. The amplitude imbalance is <0.5 dB for the RF signal frequencies above 14.5 GHz.

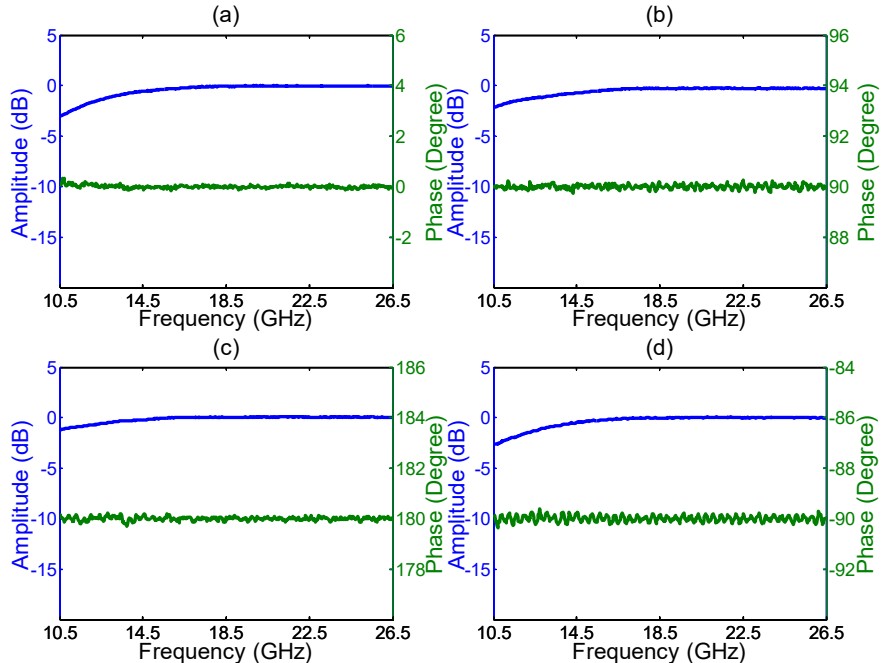

**Figure 13.** The measured amplitude and phase response of the 4-way RF hybrid splitter (**a**) 0°, (**b**) 90°, (**c**) 180° and (**d**) 270° (−90°) channel.

The 90° hybrid coupler and the 4-way RF hybrid splitter with a low amplitude imbalance and phase deviation described above have a fixed and standard characteristic. A microwave hybrid coupler with a tunable coupling ratio and phase difference can be realized by utilizing the programmable amplitude and phase control functions in an LCOS based optical processor. Figure 14 shows the structure of a broadband tunable hybrid coupler. It has two input ports (Port 1 and 3) and two output ports (Port 2 and 4) [60]. This enables two RF signals to be combined into one output port and one RF signal to be split into two output ports. The hybrid coupler shown in Figure 14 consists of two optical sources, an optical phase modulator, an LCOS based optical processor and two photodetectors, which has the same structure as the LCOS based microwave photonic device shown in Figure 2. The novelty of the structure shown in Figure 14 is that the optical phase modulator is operated in both directions. Note that a conventional optical phase modulator has an RF input port, which is located next to the phase modulator optical input port in a traveling wave configuration. In this case, the RF signal travels in the same direction as the CW light inside the phase modulator. The RF signal is terminated by a 50 Ω terminator in the phase modulator RF output port. An RF phase modulated optical signal is generated at the output of the phase modulator. In the photonics-based microwave hybrid coupler, two different-wavelength CW light ($\lambda_1$ and $\lambda_2$) launch into the optical

phase modulator in an opposite direction via optical circulators. Due to the velocity mismatch effect in electro-optic modulators, the modulation efficiency for the counter-propagating microwave and optical signal is much lower than the co-propagating case at high frequencies [61]. Hence, the effect of the counter-propagating microwave signal can be neglected at high frequencies. Therefore, the light with $\lambda_1$ and $\lambda_2$ wavelength are only modulated by the RF signals at the hybrid coupler Port 1 and 3 respectively. Note that the two different-wavelength counter-propagating phase modulated optical signals are generated by the same optical phase modulator. This eliminates the amplitude and phase imbalance between the two hybrid coupler input ports. This is important for the hybrid coupler to be used as a power combiner to combine two RF signals with the desired coupling ratio and phase difference remain the same throughout the coupler bandwidth. The two different-wavelength RF phase modulated optical signals are combined at a 2-to-1 50:50 optical coupler before entering an LCOS based optical processor. The processed optical signals at the two LCOS based optical processor output ports are detected by two photodetectors. Coupling ratio and phase tuning of the photonics-based microwave hybrid coupler shown in Figure 14 can be obtained via the programmable amplitude and phase control function in the LCOS based optical processor.

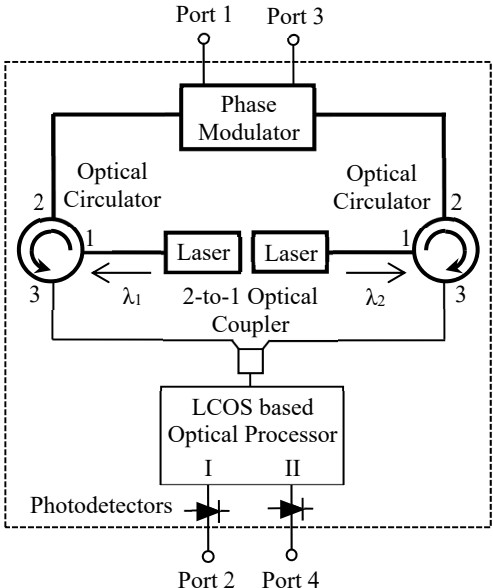

**Figure 14.** The structure of the photonics-based microwave hybrid coupler. The bold line represents the polarization maintaining components.

Figure 15 shows the measured amplitude and phase responses at Port 2 and 4 of the hybrid coupler. The results show that the phase difference between the two outputs can be continuously tuned from $-180°$ to $180°$. All the phase responses are flat with less than 1.3° deviation from the desired values over the 6–16 GHz frequency range. Such small frequency-dependent phase response performance cannot be achieved by all reported phase-tunable microwave hybrid couplers. The hybrid coupler amplitude responses for different phase shifts are also flat at high frequencies. The amplitude imbalance between the two hybrid coupler output ports ($S_{21}$–$S_{41}$) was investigated. It was found that the two output ports have less than 1.3 dB amplitude imbalance for various phase differences in the 6–16 GHz frequency range. The hybrid coupler also has an excellent input amplitude imbalance performance due to the two hybrid coupler input ports are the two RF ports of the optical phase modulator and hence they have the same frequency response characteristic. Experimental results also demonstrate that the hybrid coupler is capable of operating as a power combiner and as a splitter simultaneously. The photonics-based microwave hybrid coupler should find applications in the next generation wireless communication systems [7] and electronic warfare systems.

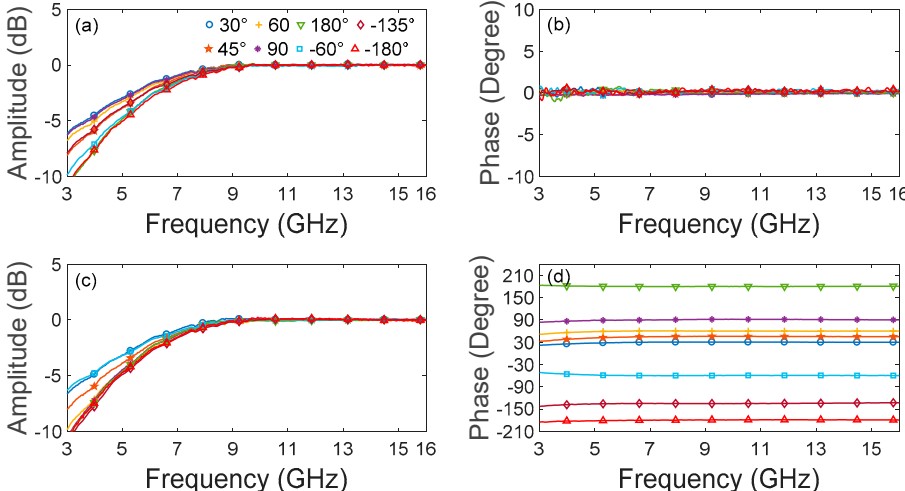

**Figure 15.** The measured (**a**) S$_{21}$ amplitude response, (**b**) S$_{21}$ phase response, (**c**) S$_{41}$ amplitude response and (**d**) S$_{41}$ phase response of the photonics-based microwave hybrid coupler designed to have a 3 dB coupling ratio and a tunable phase. Port 3 of the hybrid coupler was terminated by a 50 Ω terminator.

## 6. Multi-Function Microwave Photonic Signal Processors for Phased Array Antenna Systems

A true time delay beam steering technique implemented using true time delay elements or time shifters can avoid the beam squint problem in wideband phased array antenna systems [62]. Unlike a phase shifter, which provides a tunable frequency-independent phase shift, a time shifter has a linear phase response with a tunable slope. Widely and continuously tunable time delays are needed to obtain a wide beam steering range. For example, considering a phased array antenna system with a half wavelength antenna separation operates at 12 GHz (X-band). The required time delay between adjacent elements in order to obtain a beam direction of 60° left of the boresight is 36.1 ps. This implies that a 288.8 ps time delay is needed for a 9 × 9 phased array to achieve a ±60° beam steering range. It is difficult for an electronic device to realize such a large time delay in a continuously tunable mode. A conventional method to implement true time delay beam steering is by using a series of switches. In this case, the signal time delay is controlled in a discrete fashion. Many switches are needed for realizing a high-resolution phase shift resulting in a bulky and complex structure. Combining true time delay beam steering with phase-based beam steering using phase shifters can simplify the implementation of phased array [63]. In this case, the total array is partitioned into sub-arrays. Time shifting is applied in feeding the sub-arrays with phase shifting in the excitation of the individual element. Squinting in the sub-arrays is negligible due to their reduced size.

Many microwave photonic techniques for realizing true time delay [64–66] or phase shifting operation, have been reported. The phase response profile of an LCOS based optical processor can be designed to have a tunable linear phase slope to obtain the time-shifting operation. Additionally, the power splitting function in the commercial LCOS based optical processor enables a single-wavelength laser source to generate multiple independently tunable true time delays suitable for phased array antenna systems. However, a commercial LCOS based optical processor can only realize a limited amount of time shift ranged from −32 ps to +32 ps [65] so that a 0°–360° phase shift cannot be obtained for microwave signals with frequencies below 30 GHz. This limits the beam steering range. This problem can be solved by a new LCOS based time and phase shifter, which has a structure as shown in Figure 2. CW light from a single-wavelength laser source is phase modulated by an RF signal generating a pair of anti-phase RF modulation sidebands together with an optical carrier. The phase response profile of the LCOS based optical processor is designed to introduce a linear phase shift to the left sideband and a constant phase shift to the right sideband. The left sideband beat with the optical carrier at the photodetector generates an RF signal with a linear slope phase response. The right sideband beat with the optical carrier at the photodetector generates an RF signal with a

constant phase shift independent to the RF signal frequency. Therefore the overall phase response of the output RF signal has a linear slope plus a constant phase shift.

Unlike the previously reported LCOS based time shifter [65], there is no removal of an RF modulation sideband in the new LCOS based time and phase shifter. Hence the proposed structure has a high output RF signal power, which leads to a high SNR performance. It is worth noting that the output RF signal amplitude can be controlled via the amplitude control function in the LCOS based optical processor. This shows the three fundamental operations required in phased array antenna systems, i.e., that the time shift, phase shift and amplitude control, can be implemented in a single unit. Using the power splitting function in a commercial LCOS based optical processor in conjunction with a wavelength division multiplexing (WDM) technique, the new multi-function microwave photonic signal processor can generate a large number of RF signals with tunable amplitudes, and time and phase shifts. The new LCOS based time and phase shifter was set up experimentally. The amplitude and phase response profile of the LCOS based optical processor were first designed to demonstrate a tunable time-shifting operation. Figure 16 shows the measured phase and amplitude response of the new time and phase shifter based on LCOS technology. The results demonstrate a tunable true time delay with a less than 3 dB amplitude variation over a very wide 10 to 40 GHz frequency range. Around 6 dB output RF signal power improvement was obtained across the operating frequency range compared with the previously reported LCOS based time shifter using SSB modulation. The LCOS based optical processor amplitude and phase response profile were then designed to obtain a 10 ps time delay and a tunable 0°–360° phase shift. The measured phase and amplitude response are shown in Figure 17. It can be seen that the four LCOS based time and phase shifter phase responses have the same slope but different phase shifts of 0°, 90°, 180° and 270° at 10 GHz. The amplitude variation is less than 2.5 dB over the 10 to 40 GHz frequency range. The results demonstrate the simultaneous realization of both time and phase shifting operations in a single unit.

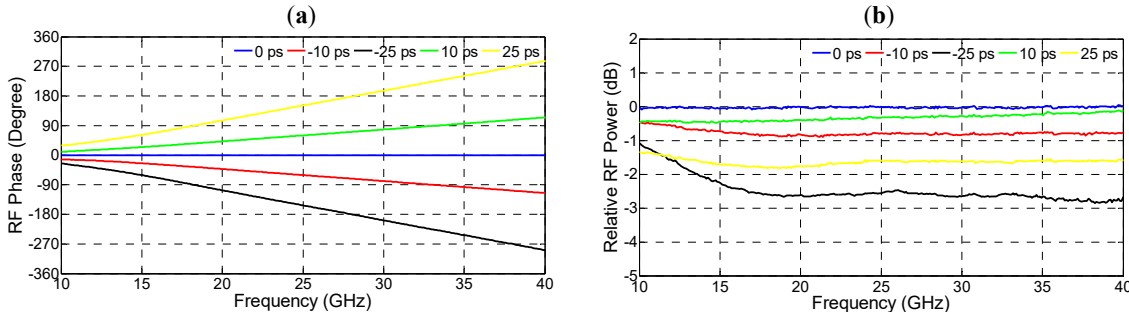

**Figure 16.** (**a**) The measured phase and (**b**) amplitude response of the new multi-function microwave photonic signal processor designed to realize a tunable true time delay operation.

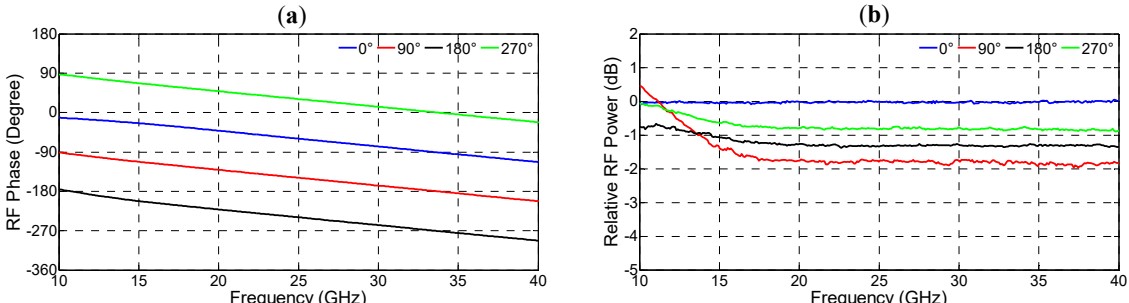

**Figure 17.** (**a**) The measured phase and (**b**) amplitude response of the new multi-function microwave photonic signal processor designed to simultaneously realize a fixed 10 ps time delay and a tunable phase shifting operation.

## 7. Conclusions

Microwave photonics provides a solution to overcome the limitations in electronic devices. Devices implemented using microwave photonic techniques can process microwave and millimeter wave signals over a wide frequency range. They can be integrated with fiber optic communication systems and are immune to EMI. Microwave photonic devices based on LCOS technology offer additional advantages of high reconfigurability, parallel signal processing and robust performance. Recent works on using LCOS based optical processors to implement widely tunable high-resolution notch filters, continuously tunable 0°–360° phase shifters, and low amplitude and phase imbalance hybrid couplers have been described. Additionally, a new LCOS based microwave photonic device that can perform multiple RF signal time shifting, phase shifting and amplitude controlling operations at the same time has been presented. It is capable to generate multiple output RF signals with arbitrary amplitude and phase shifts. Experimental results have demonstrated simultaneous time and phase shifting operations over an extremely wide bandwidth of 10 to 40 GHz. Around 6 dB improvement in the output RF signal power across the operating frequency range in comparison to the previously reported LCOS based time shifter using SSB modulation has also been demonstrated. The LCOS based microwave photonic devices presented in this paper should find applications in radar, electronic warfare, and communication.

**Author Contributions:** Writing—original draft preparation, formal analysis and investigation, R.Z.; writing—review and editing, E.H.W.C. and X.W.; project administration and supervision: X.W., X.F., and B.-O.G.

**Funding:** This research was funded by the National Natural Science Foundation of China (NSFC, Beijing, China), (No. 61501205, 61771221).

**Conflicts of Interest:** The authors declare no conflict of interest.

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
