# Peer review of "Microwave Photonic Devices Based on Liquid Crystal on Silicon Technology"

_applsci, doi:10.3390/app9020260_

Round 1

Reviewer 1 Report

Great works! The paper well descirbed the operation principle, functions and important specifications of a LCOS based optical processor. It is suggested that it would be much better to the readers, comparing to the other photonic devices such as Bragg gratings in terms of performance, cost, and reliability in commercially practical application including future trends.

Reviewer 2 Report

Dear Authors,

In the manuscript, the authors provide the development of different microwave photonic devices, such as microwave photonic notch filter, phase shifter and couplers, based on an optoelectronics scheme that uses a Liquid Crystal on Silicon optical processor for achieving the desired behavior.

In the following, I have written about some aspect of the manuscript that should be clarified:

Related to the optical elements (Lasers, modulators, photodetectors) there are not any information about type of elements, wavelengths, characteristics.... This refers to all the described devices.

In this way, the authors say the bandwidth limitations of some of the proposals are due to the phase modulator. Why not other devices? It could be more interesting to give more detail about this assertion.

It is suggested to improve the Figures 4 and 10. It is quite difficult to read the axes.

In page 5, line 208, the authors say: "This demonstrates the novel all-optical microwave photonic notch filter has an excellent long term stability performance, which cannot be achieved by many reported notch filtering structures".

Also, in page 7, line 269, the authors say: "Less than 0.2 dB amplitude variation and less than 2º phase deviation was obtained over the measurement period of almost 5 hours. This demonstrates that the LCOS based microwave photonic phase shifter has an excellent long term stability performance".

In this way, it could be interesting to include any study or comparison about long term stability with other designs.

In page 8, line 282, the authors indicate the problem of the Signal to Noise Ratio degradation due to the fact of one RF sideband removal. There is not any information about this problem in the previous designs. Neither there is any information about Insertion Loss and other interesting parameters in the case of being necessary.

In page 11, Figure 14, the FD-OP it is supposed to be the LCOS, isn't it? FD-OP is not indicated in the text.

In the same figure, the 2 input ports optical phase modulator requires a better explanation.

Best regards.

Reviewer 3 Report

This is a review article and the figures are reused from their work.

I think the operation principle should be described in more detail even if those are written in the reference paper.

Round 2

Reviewer 3 Report

I think that the authors revised their manuscript appropriately. I recommend this review paper to be published.